# GerenciaVida: Validity Evidence of a Mobile Application for Suicide Behavior Management

**DOI:** 10.3390/ijerph22071115

**Published:** 2025-07-15

**Authors:** Daniel de Macêdo Rocha, Aline Costa de Oliveira, Sandra Marina Gonçalves Bezerra, Laelson Rochelle Milanês Sousa, Rafael Saraiva Alves, Breno da Silva Oliveira, Iara Barbosa Ramos, Muriel Fernanda de Lima, Renata Karina Reis, Lídya Tolstenko Nogueira

**Affiliations:** 1Department of Nursing, Federal University of Mato Grosso do Sul, Coxim 79400-000, Brazil; rafael_s@ufms.br (R.S.A.); oliveira_breno@ufms.br (B.d.S.O.); iara.ramos@ufms.br (I.B.R.); muriel.lima@ufms.br (M.F.d.L.); 2Ribeirão Preto College of Nursing, University of São Paulo, Ribeirão Preto 14040-902, Brazil; rkreis@eerp.usp.br; 3Department of Nursing, Federal University of Piauí, Teresina 64049-550, Brazil; alinecosta.1@hotmail.com (A.C.d.O.); sandramarina@ccs.uespi.br (S.M.G.B.); lidyatn@gmail.com (L.T.N.); 4Nursing Course, State University of Maranhão, Coroatá 65665-000, Brazil

**Keywords:** suicide, suicide prevention, technology, smartphone, mobile applications, validation study

## Abstract

Technology-based strategies for the prevention and management of suicidal behavior are widely referenced for identifying vulnerable groups and for supporting clinical reasoning, decision-making, and appropriate referrals. In this study, we estimated the interface and content validity evidence of an interactive mobile application developed for managing suicidal behavior. This methodological study employed psychometric parameters to evaluate the content and interface of the mobile application, following five action phases: analysis, design, development, implementation, and evaluation. A total of 27 healthcare professionals participated, selected by convenience sampling, all working within the Psychosocial Care Network across different regions of Brazil. Data were collected using an electronic form, the Delphi technique for evaluation rounds, and a Likert scale to achieve consensus. The validity analysis was based on a Content Validity Index (CVI) equal to or greater than 0.80. The results showed that GerenciaVida, a technology developed for healthcare workers—regardless of their level of care or professional category—can be used to screen for suicide risk in the general population and indicate preventive alternatives. The app demonstrated satisfactory indicators of content validity (0.974) and interface validity (0.963), reflecting clarity (0.925), objectivity (1.00), adequacy (0.925), coherence (0.962), accuracy (0.962), and clinical relevance (1.00). The development path of this mobile application provided scientific, technological, and operational support, establishing it as an innovative care tool. It consolidates valid evidence that supports the identification, risk classification, and prevention of suicidal behavior in various healthcare contexts.

## 1. Introduction

Suicidal behavior constitutes an important indicator of morbidity and mortality within the global population, characterized as a complex, predictable, multidimensional, and universal phenomenon [1,2]. This manifestation can vary in severity levels, encompassing suicidal ideation, planning, attempts, and death by suicide, and is expressed through deliberate and intentional acts of self-harm with a strong expectation of a fatal outcome [3]. A growing trend in suicidal behavior indicators has been observed in low-, middle-, and high-income countries following the COVID-19 pandemic [4,5]. The effects of this scenario reflect a high mental health burden, with increased psychological distress, symptoms of stress, anxiety, depression, and heightened suicide risk [6,7].

Global estimates projected by the World Health Organization (WHO) characterize suicidal behavior as a significant public health problem [8]. In 2021, the global suicide mortality rate corresponded to 746,000 deaths [9]. Epidemiological variations are described according to the population’s sociodemographic, social, economic, and health conditions [8,9]. Brazil holds one of the highest suicide prevalence rates in the world and remains among the ten countries with the highest absolute number of cases. Although public policies for suicide prevention are structured in the country, initiatives for early recognition, screening, risk classification, and necessary referrals remain limited, incipient, and often neglected [10].

In this context, the development of technology-based strategies gains prominence for the implementation of mental health care, prevention, and management of suicidal behavior [11,12]. In the literature, mobile applications are widely referenced as tools that facilitate the systematization of healthcare work and show potential for use across different levels of care. Moreover, they can consolidate evidence supporting the identification of vulnerable groups, as well as the determinants and predictors of risk behavior [13,14].

Easy access, wide availability, and data storage and sharing capabilities—common features of mobile technologies—support clinical reasoning and guide decision-making and care planning [13,14]. Furthermore, the incorporation of these technologies in mental health may lead to improvements in communication indicators, risk management, remote monitoring of clinical conditions, and the enhancement of professional knowledge, skills, and competencies [15].

Although the development of mobile applications for mental health management has increased in recent years, their content and outcomes are primarily focused on measuring quality of life and assessing the presence, intensity, and associated factors of stress, depression, and anxiety in specific groups. These tools still demonstrate low sensitivity and specificity for predicting suicide risk [16,17].

Given the magnitude of the problem, the positive effects of technology on mental health practices, the prevention possibilities through early identification, and the care limitations that contribute to underreporting, this study aimed to assess the evidence for content and interface validity of an interactive mobile application developed for the management of suicidal behavior.

## 2. Materials and Methods

### 2.1. Study Design

This is a methodological study based on the Contextualized Instructional Design Model (DIC) and structured in five action phases: analysis, design, development, implementation, and evaluation [18]. This article followed the CREDES checklist in the Equator Network to ensure methodological quality [19].
**Phase 1—Analysis**

The analysis phase involved an exploratory stage that included content planning, defining the target audience and preliminary objectives, technological infrastructure, and care process diagrams.

An integrative literature review was conducted to map key concepts and technologies developed in this field, analyze the scope, extent, and methodological quality of the included studies, synthesize the results, and identify existing gaps [17]. The six steps proposed by Whittemore and Knafl (2005) were followed: formulation of the research question; literature search and sampling; definition of information to be extracted; critical appraisal of studies; interpretation of results; knowledge synthesis; and presentation of the review [20].

Twelve primary studies focused on the construction, validation, and evaluation of technologies capable of measuring suicide risk were selected from the following databases: MEDLINE via PUBMED, Web of Science, CINAHL, SCOPUS, LILACS, and BDENF via the Virtual Health Library (VHL) [17].

Public policies related to emergency care, humanization, and mental health, as well as referral manuals from the Psychosocial Care Network (RAPS), also supported the instructional planning. Guidelines, strategies, and clinical recommendations for appropriate referral based on suicide risk were incorporated [21,22].

Additionally, a cross-sectional analytical survey was conducted at a psychiatric emergency department in Teresina, Piauí, Brazil. This aimed to characterize the sociodemographic, clinical, and therapeutic profile of 130 adult patients seen after suicidal ideation, planning, or attempt, as well as to understand factors associated with risk behavior and care needs [23].
**Phase 2—Design**

This phase considered aesthetic aspects of the technology and visual identity, including symbols, color palette, and typography. Functionalities, navigation elements, animations, illustrations, infographics, forms, textual organization, tables, links, and the system of labels and titles were also defined.

Given that suicidal behavior is a multidimensional phenomenon, the assessment modules, design, and navigation interfaces were developed by a team consisting of a nurse, psychiatrist, social worker, and information technologist, all with specialization and clinical or academic expertise in the field.
**Phase 3—Development**

The development involved creating the interface, visual resources, logical organization, and hierarchical content structure. This systematic planning aimed to define the layout of consultative elements and their aesthetic–formal arrangement, ensuring a user-friendly technological model with accessible features.

During this phase, the Suicide Risk Scale was adopted for identifying suicidal behavior. This public-domain tool is validated for the Brazilian context and presents reliable evidence for classifying patients as low, moderate, or high risk, guiding screening and preventive measures [24].

Additionally, the Hamilton Depression Rating Scale (HAM-D) and the Beck Anxiety Inventory (BAI) were integrated into the technological solution due to the association between anxiety, depression, and suicide risk. The HAM-D assesses 17 items related to emotional, cognitive, and physical aspects of depression, while the BAI evaluates 21 items concerning somatic and subjective manifestations of anxiety. Both instruments are psychometrically robust, validated for use in Brazil, and widely incorporated into clinical practice for emotional assessment and intervention monitoring [25,26].
**Phase 4—Implementation**

Initially, an invitation letter and an Informed Consent Form (ICF) were sent to participants. Upon consent, the developed technology, along with installation and usage guidelines and evaluation instruments for content and interface, was made available via Google Drive^®^.
**Phase 5—Evaluation**

The validation process for content and interface was conducted using the Delphi technique [19], which seeks to achieve consensus among experts on a specific topic through multiple evaluation rounds. The expert panel consisted of professionals working in the study’s field of interest, who demonstrated motivation to critically reflect on the topic and its applicability in professional practice [27].

### 2.2. Study Period

The study operational phases were as follows: integrative literature review (September to December 2019); cross-sectional survey (December 2020 to March 2021); and content validation (January to May 2022).

### 2.3. Population, Selection Criteria, and Sample

To compose the expert panel, 27 health professionals were intentionally selected, all working in different services within the Psychosocial Care Network (RAPS). The sample size followed recommendations in the literature, which suggest a minimum of seven participants without requiring a statistical representativeness calculation [28].

The selection process involved consulting the researchers’ résumés on the Lattes Platform of the National Council for Scientific and Technological Development (CNPq), assessing technical qualifications and the number of relevant attributes. Experts were included if they scored five or more points based on Fehring’s criteria: doctorate in the area of interest; master’s degree; thesis or dissertation focused on mental health; at least one year of academic or professional experience with the target population; published scientific production in indexed journals in the area; and/or a specialist degree in mental health [29].

### 2.4. Data Collection

The validation process occurred in two rounds, with a 15-day response window. A validated and adapted questionnaire with three dimensions was used: 1—Sociodemographic, occupational, and educational characterization of the experts (sex, origin, education level, professional category, and area of expertise); 2—Content evaluation, assessing clarity, adequacy, use of evidence, coherence, relevance, content updating, logical organization, accuracy, and objectivity; 3—Interface evaluation, measuring the technological interface, usability resources, limitations, and the naming of the mobile application. Interface and content were rated using a five-point Likert scale: 1—No opinion; 2—Strongly disagree; 3—Disagree; 4—Agree; 5—Strongly agree [30]. An open field was also provided for suggestions regarding adjustments, reformulations, or additions.

### 2.5. Data Analysis and Processing

Data were double-entered into Excel (Office 2016) and exported to Statistical Package for the Social Sciences (SPSS) version 26. Sociodemographic, occupational, and educational data were analyzed using descriptive statistics, including mean, standard deviation, maximum and minimum values, and absolute and relative frequencies.

The validation process involved calculating the Content Validity Index (CVI) to measure the proportion or percentage of agreement among experts. The score assigned by each expert for each item and dimension was summed and divided by the total number of participants. Psychometric indicators with a CVI equal to or greater than 0.80 and an overall mean higher than 0.90 were considered satisfactory. Suggested revisions were incorporated into the final version of the technological tool.

### 2.6. Ethical Considerations

This study was approved by the Research Ethics Committee of the Federal University of Piauí, under approval number 4.444.303. Participation was voluntary and conditional upon signing the Informed Consent Form.

## 3. Results

### 3.1. Technological Development

The evidence included in the integrative review highlights valid, sensitive, specific, and objective care alternatives capable of screening, identifying, and classifying the severity of suicidal behavior, as well as expanding access to specialized services and improving knowledge, self-confidence, and professional performance for clinical decision-making, care management, and the implementation of suicide prevention and control strategies. Software, computational algorithms, and learning tools were identified which, despite their potential, presented clinical, scientific, and psychometric limitations. The absence of technological resources with validity evidence for use in Brazil stood out as the main unmet need and represented a significant gap in the country’s suicide prevention efforts.

The technologies already developed are also limited in recognizing the different manifestations of suicidal behavior and the impacts of the COVID-19 pandemic on suicide risk. Furthermore, the combination of different suicide prevention strategies within a single technological resource, the emphasis on protective factors in the general population, and the potential for use by various professional categories and in different healthcare settings are still underexplored in the available technological resources (Table 1).

The first version of the technology was named GerenciaSui, developed in Portuguese and made available in a hybrid format for use on both Android and iOS platforms. It was designed for the identification and classification of suicidal behavior, implementation of preventive measures, and appropriate care referrals.

The target audience consisted of healthcare professionals, regardless of their level of care or professional category, who were required to register with their professional credentials to access the functionalities of the mobile application.

The initial content included patient identification data such as name, age, gender, and designated caregiver. Suicide risk assessment was performed based on six items measuring the presence of suicidal behavior in the past month, along with the presence and intensity of anxiety and depression symptoms. Factors identified in the cross-sectional survey were also incorporated, including history of mental disorders or previous suicide attempts, use of psychoactive substances, income loss, and marital instability.

Following clinical assessment, the platform estimates suicide risk (low, medium, or high) and recommends preventive approaches and necessary referrals. The care pathways are based on the identified risk level and the operational structure of the services that make up the Psychosocial Care Network (RAPS) in Brazil.

The preventive interventions include providing support, family involvement, behavioral monitoring and surveillance, mandatory reporting in cases of self-harm, and referrals for specialized assessment in cases of suicide attempts, ideation, or planning. Finally, a descriptive report can be generated, with the potential for data storage and sharing, containing information on the suicide risk level, evidence of anxiety and depression, associated conditions, proposed interventions, and recommended care actions.

### 3.2. Content and Interface Validation Using the Delphi Technique

#### 3.2.1. Characterization of Experts

The validation process involved a committee of 27 experts, composed of 12 nurses (44.4%), 6 physicians (20.7%), 4 social workers (14.8%), and 4 psychologists (14.8%). These professionals were involved in education (9–33.3%), research (6–22.2%), and clinical practice (12–44.4%), mainly within various psychosocial care network services and specialized suicide prevention centers.

Most participants were female (66.7%), from the Northeast region of Brazil (15–55.6%), with specialization (16–59.3%), master’s degrees (26–96.3%), and doctoral degrees (14–51.9%) in the field of interest. The average professional experience was 6.4 years (SD ± 3.9).

#### 3.2.2. Assessment Rounds

Two evaluation rounds were conducted with the same group of experts. In the first round, despite already demonstrating evidence of validity with a global Content Validity Index (CVI) of 0.908, the panel suggested adjustments that improved indicators related to clarity, alignment with proposed objectives, coherence, up-to-date information, logical organization, accuracy, and objectivity. In both rounds, the global CVI and the CVI across all categories remained above 80%, which is considered satisfactory according to the adopted standards (Table 2).

The interface also demonstrated validity evidence in the first round of evaluation (global CVI of 0.888). The revisions made were considered by the experts as essential elements to ensure the easy use of the technology, as well as to minimize accessibility and comprehension limitations for the target audience. Table 3 shows the scores demonstrating validity in terms of presentation and usability.

Table 4 presents the adjustment requests and revisions resulting from the first round of evaluation. The revisions involved both interface and content, including graphical redesign, adaptation of the application name to GerenciaVida, adjustment of technical terms, inclusion of additional vulnerability factors related to suicidal behavior, and revision of referral recommendations according to the identified risk level.

Figure 1 presents the main functionalities of the GerenciaVida mobile application, as well as its key interfaces, which include system access, suicide risk assessment, monitoring of anxiety and depression symptoms, tracking current health status, and generating a report after the consultation.

## 4. Discussion

The complex nature of suicidal behavior and the limitations of existing technologies reinforce the need for new tools that assess individual, psychological, social, and environmental markers in the Brazilian population. This study focused on the development and validation of a mobile application with evidence-based capacity to track risk, implement prevention measures, and facilitate referrals within Brazil’s Psychosocial Care Network. The content foundation was based on previous studies that identified availability, strengths, and gaps in existing technologies; prevention guidelines; prevalence coefficients; and pandemic-related risk factors during the COVID-19 era [17,23].

Interface and content validation constituted a fundamental stage in the technological development process [18,19,27]. Subjective evaluations were carried out by a thematic-area expert committee to ensure the clarity, relevance, and appropriateness of the proposed items for the target audience. Although the specialist panel was regionally concentrated due to convenience sampling, a significant portion of participants were from other regions of Brazil. The sample size exceeded minimum requirements for validation studies and encompassed multiple professional contexts [28]. These factors brought together diverse perspectives on screening, prevention, and management of suicidal behavior and suggest the technology’s applicability across territories with distinct socioeconomic, cultural, and structural dynamics.

The evidence for interface and content validity of the mobile app was deemed adequate, demonstrating scientific, technological, and operational support. Agreement coefficients also confirmed the technology’s clarity, alignment with its objectives, coherence, contemporaneity, logical organization, precision, and objectivity.

### 4.1. Practical Implications of Using GerenciaVida

Despite the growth in the development of mobile technologies aimed at managing mental health-related conditions, significant limitations have been identified, which may pose challenges to the screening and provision of rapid and effective responses for the prevention of suicidal behavior in the Brazilian population [17].

We developed the GerenciaVida mobile application to support the screening, risk classification, referral, and prevention of suicidal behavior in Brazil, overcoming limitations related to the lack of a validated tool that considers the healthcare network and regional variations across the country. Its features, designed for the early identification of different manifestations of suicidal behavior, the integration of various prevention strategies, and the coordination with mental health services, constitute a technological innovation and highlight its potential for incorporation into real-life clinical practice.

GerenciaVida expresses the potential for use in Primary Health Care and home-based contexts, reflecting a key characteristic of mobile technologies. Expanding the scope of screening actions beyond the physical limits of health services is a strategic move for suicide prevention and control in Brazil. Primary Health Care and home care play a crucial role in managing risk factors and enabling timely interventions for individuals who do not spontaneously seek health services [31,32,33].

In Brazil, there is a high prevalence of individuals in conditions of psychosocial vulnerability who do not access formal health services for proper assessment and timely intervention. Social isolation, anxiety and depression symptoms, and stigma experienced by patients reduce their pursuit of mental health care and substantially increase the risk of suicide [34]. In this context, mobile technologies become powerful tools to track individuals in vulnerable situations within their communities, assess the severity of the episode, refer them to specialized services, implement prevention strategies, and strengthen social bonds and support networks [31].

GerenciaVida supports suicide behavior screening by various professional categories, representing a promising approach to enhance the early detection of vulnerable cases and strengthen prevention efforts, particularly in contexts with limited access to specialized services. In the Brazilian healthcare system, home care is provided by multiprofessional teams working directly in communities, responding to the needs of populations that often do not access conventional services due to geographic, social, clinical, or psychological barriers [35]. Providing a mobile, interactive, validated, and accessible tool that enables screening regardless of the professional category contributes to expanding the reach of prevention actions and enhancing the quality of care across different points in the care network.

Another practical implication involves the systematized screening of different manifestations of suicidal behavior and risk stratification. Early identification of suicidal ideation and planning is essential to prevent more severe outcomes. This approach enables targeted interventions and helps reduce the risk of suicide by increasing care opportunities before critical episodes occur [36].

Evidence from the cross-sectional survey contributed to recognizing the impact of the COVID-19 pandemic on suicide risk. Individual, economic, social, biological, and health-related predictors were incorporated into the initial design of the technology [23]. In addition to a history of mental disorders, previous suicide attempts, and the use of psychoactive substances, income loss and relationship instability formed the basis for content development. During the pandemic, financial hardship, income loss, and marital crises increasingly contributed to the emergence or intensification of psychopathological comorbidities that can lead to suicidal behavior [37].

It is also worth emphasizing that integrating multiple suicide prevention strategies into a single technological tool constitutes a meaningful advancement in tackling this severe public health problem. GerenciaVida incorporates features such as monitoring, automated guidance, and data storage and sharing. This multidimensional approach was developed based on internationally recognized guidelines and global policy frameworks. The WHO’s LIVE LIFE implementation guide (2021) enabled the development of an innovative, robust, and adaptable technological resource for suicide risk screening across diverse contexts [38].

### 4.2. Challenges and Barriers to Technological Implementation in the Public Health System

Despite its implications for clinical practice, implementing digital technologies such as GerenciaVida in public health systems faces sociotechnical, ethical, and institutional challenges that must be systematically addressed. The heterogeneity of technological infrastructure across health services, and the dependency on internet access and the availability of mobile devices, may reduce the technology’s performance within the Brazilian public health system. Sociotechnical barriers are also evident and reflect the high rates of digital illiteracy among healthcare professionals in Brazil. Difficulties in handling mobile devices, digital platforms, and computerized systems, along with resistance to changes in work processes, are significant challenges, especially considering regional inequalities, operations in vulnerable territories, and the overload of healthcare services [39,40].

Additional concerns involve ethical and legal aspects related to data storage, privacy, and security, particularly in dealing with sensitive conditions such as suicide risk among vulnerable populations. The use of applications like GerenciaVida requires strict attention to ensure data protection, user acceptability, and technological effectiveness. This perspective broadens ethical discussions around the incorporation of mobile technologies in mental health and emphasizes that dilemmas related to data privacy, the intensification of medicalization in everyday life, digital exclusion, and subjective assessment mediated by digital platforms must be addressed [41,42].

At the institutional level, barriers are represented by the lack of guidelines and regulations governing the use of technology in mental health and the limited coordination among levels of health management to ensure continuous investment in innovation, technology, and professional training [43].

Although barriers are evident, offering a hybrid-format technology compatible with Android and iOS platforms, adapted to the demands imposed by the pandemic context and validated by experts from different regions of Brazil, demonstrates its relevance, accessibility, and innovation. This is especially true when the goal is to enhance the reach, effectiveness, and responsiveness of the healthcare system in suicide prevention. The potential of GerenciaVida to track markers of vulnerability related to suicidal behavior in different healthcare contexts reinforces its feasibility for incorporation into primary care. It is important to emphasize that suicide risk is not static and can be influenced by various factors; and that, despite the technological contributions to screening, the mobile application does not replace anamnesis or clinical assessment.

This study was conducted using a rigorous methodology, and although it provides strong evidence of interface and content validity, it is important to acknowledge its limitations. The estimated psychometric properties do not express construct validity, criterion validity, empirical validation with end users, usability indicators, or reliability measures. Furthermore, clinical impacts in real-world healthcare settings were not measured in this investigation. Recognizing these gaps allows for the development of future validation plans, the expansion of geographic representativeness within the target population, and the design of longitudinal studies to assess usability, transparency, and reproducibility of digital interventions, as well as clinical effectiveness, referral success rates, and prevention of critical episodes in different regional contexts of Brazil.

## 5. Conclusions

The trajectory undertaken for the development of the GerenciaVida mobile application provided scientific, technological, and operational support, establishing it as an innovative care resource by bringing together evidence that supports the identification, screening, risk classification, and prevention of the various manifestations of suicidal behavior. The interface and content validity indicators were considered satisfactory, indicating clarity, alignment with the proposed objectives, coherence, up-to-date information, logical organization, accuracy, and objectivity of the developed technology.

The participation of experts from various professional fields, who work within the psychosocial care network and specialized suicide prevention centers, demonstrates the high potential for application in different contexts and levels of healthcare. The structure of the application was based on the identification and risk classification of suicidal behavior, as well as the assessment of anxiety and depression levels, the implementation of preventive measures, and the provision of care according to the identified risk. Additionally, the possibilities of storing and sharing health information are noteworthy. Further studies are needed to assess the technological impact on professional performance regarding suicide screening, monitoring, and prevention, as well as on access to specialized services and notification indicators. This is crucial, considering the app’s potential to support the implementation of public policies for mental health protection and the promotion of life.

## Figures and Tables

**Figure 1 ijerph-22-01115-f001:**
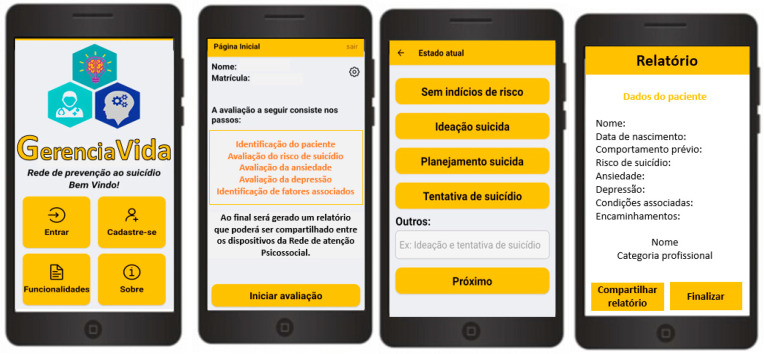
Navigation interfaces of the GerenciaVida mobile application.

**Table 1 ijerph-22-01115-t001:** Summary of care technologies developed for managing the risk of suicidal behavior, Teresina, Piauí, Brazil, 2020.

Technology	Applicability	Objective	Limitation
HelPath (Computational Platform)	Screening and monitoring	Support triage and direct to specialized services based on risk level.	Did not evaluate outcomes of the technology’s implementation.
App-assisted treatment	Monitoring	Assess the impact of a mobile app on depression and suicide risk.	Results were generalized to the adult population.
Natural Language Processing	Screening and data navigation	Estimate suicide risk through combined analysis of social media, demographic, and mental health data.	Ethical and privacy implications. Requires individual consent. No evidence of effectiveness across cultures.
Linehan Suicide Safety Net	Monitoring and data navigation	Assess, manage, and document suicide risk.	Only evaluated medical professionals’ performance, not other key professional categories involved in risk assessment.
e-PASS Suicidal Ideation Detector (eSID)	Screening and data navigation	Identify suicidal ideation in Primary Health Care.	Focused only on identifying suicidal ideation.
Computational Algorithm	Screening and data navigation	Identify markers of suicidal thinking in linguistic and acoustic characteristics (dynamics, frequency, vocal quality).	Technological accuracy may decrease when applied across different navigation channels.
Ecological Momentary Assessment	Screening and monitoring	Classify and recognize risk situations related to mood, self-harm, environment, and social context.	Generalization of results to female population.
Ecological Momentary Assessment	Screening and monitoring	Predict individual changes, clinical factors, and risk states.	Allows assessment in only one condition related to suicidal behavior (ideation).
Software—Lifenet	Screening	Determine suicide risk in adolescents, identify predictors, and suggest interventions.	Lacks evidence of external validity and reliability.

**Table 2 ijerph-22-01115-t002:** Content validity evidence for the mobile application developed for suicide behavior management.

Content Item	First Round	Second Round
Clarity	0.851	0.925
Adequacy	0.814	0.925
Evidence-Based	1.00	1.00
Coherence	0.814	0.962
Relevance	1.00	1.00
Up-to-date Information	0.888	1.00
Logical Organization	0.925	1.00
Accuracy	0.925	0.962
Objectivity	0.962	1.00
Global CVI	0.908	0.974

**Legend**: CVI—Content Validity Index.

**Table 3 ijerph-22-01115-t003:** Interface validity of the mobile application developed for suicide behavior management.

Appearance Item	First Round	Second Round
Adequate appearance	0.963	0.963
Ease of use	0.852	0.926
Limitations	0.852	0.963
Appropriate name	0.814	1.00
Adequacy	0.963	0.963
Global CVI	0.888	0.963

**Legend**: CVI—Content Validity Index.

**Table 4 ijerph-22-01115-t004:** Revisions made to the mobile application after expert evaluation.

Recommended Adjustments	Dimension	Revision
Redesign the graphical presentation to improve attractiveness and content clarity.	Interface	Addressed
Use different shades of yellow to enhance visual comfort.	Interface	Addressed
Adapt the title to include a reference to VIDA (life) instead of SUI (suicide).	Content	Addressed
Insert the application’s purpose on the home screen.	Content	Addressed
Add mental disorder history as an associated factor, specifying the diagnosed condition if present.	Content	Addressed
Include history of mental health treatment in the assessment of associated factors.	Content	Addressed
Highlight the timeframe of symptoms in the suicide risk, anxiety, and depression assessments.	Content	Addressed
Add definitions for technical terms to support users with limited knowledge of the subject.	Content	Addressed
When investigating previous suicide attempts, specify the methods used to assess the lethality potential.	Content	Addressed
Add a tool that allows saving of responses in case users need to return to previous assessment items.	Interface	Addressed
Include patient identification variables related to gender identity, race/ethnicity, and social factors.	Content	Addressed
Indicate at the end of the assessment that the identified risk is not static, and that the app does not replace clinical evaluation and anamnesis.	Content	Addressed
Add items to assess the desire for suicidal acts and the presence of imminent risk during consultation.	Content	Addressed
Define that imminent suicide attempts require immediate intervention.	Content	Addressed
Highlight the references used for suicide risk classification.	Content	Addressed
Explain that the assessment of anxiety and depression is due to their high prevalence in patients at suicide risk.	Content	Addressed

## Data Availability

The original contributions presented in this study are included in the article. Further inquiries can be directed to the corresponding authors.

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
