# Peer review of "GerenciaVida: Validity Evidence of a Mobile Application for Suicide Behavior Management"

_ijerph, 2025, doi:10.3390/ijerph22071115_

Round 1
Reviewer 1 Report
Comments and Suggestions for Authors
First, we congratulate the authors for the quality of the work presented. The article presents a relevant, timely, and well-structured proposal on validating mobile technology for screening and managing suicidal behavior in the Brazilian context. It is a significant contribution at the intersection of mental health, technological innovation, and primary care, with potential for real impact on clinical practice.
However, some issues are identified that can be strengthened to improve the methodological quality, conceptual clarity and applicability of the study.
The validated app GerenciaVida arises as a response to the lack of effective systematic instruments for early detection of suicidal behavior. Although it is noted that other apps present low sensitivity and specificity, it would be valuable to include a review of these apps in the format of a comparative table that highlights examples of similar technologies and clarifies how GerenciaVida overcomes these limitations.
Although the article includes relevant sources such as the WHO and recent post-COVID-19 literature, the theoretical framework on the use of mobile technologies in mental health is limited. Key references on e-health, m-health, and their specific application to suicide prevention are missing. Expanding this section by incorporating critical literature that addresses the ethical dilemmas of using mobile apps in mental health, particularly among vulnerable populations, is recommended. In this regard, we recommend the work of Brazilian researcher Marilia Duque (https://mariliaduque.com/) and the research group Anthropology of Smartphones and Smart Ageing led by British anthropologist Daniel Miller (https://wwwdepts-live.ucl.ac.uk/anthropology/assa/).
The use of the DIC model and the Delphi technique is well justified regarding the methodology. The choice of psychometric tools, such as the Hamilton scale, the BAI, and a suicide risk scale validated in Brazil, reinforces the strength of the study.
While broad and addressing clinical, social, and technological aspects, the discussion contains redundancies in the introduction that weaken its analytical depth. It is recommended to restructure this section around three key axes: (1) practical implications of using the app, (2) implementation challenges within public health systems; and (3) sociotechnical or institutional barriers. Additionally, reflecting on the ethical and privacy limitations associated with apps that collect sensitive data would be relevant.
The limitations section is underdeveloped, lacking discussion on key aspects such as the absence of real-world clinical impact measurement and the lack of empirical validation with end users—patients or professionals in actual healthcare settings. It is essential to acknowledge these gaps to provide a more balanced assessment of the study's scope and reliability. As a limitation, it is recommended to include the need for longitudinal studies that assess clinical effectiveness, referral success rates, and prevention of critical episodes.
The conclusion effectively highlights the relevance and potential applicability of the tool, and its call for further research is both timely and appropriate.
Author Response
Dear Reviewer,
We appreciate your careful analysis, the issued review, and your requests for important revisions, which have led to significant improvements in the revised version. We have made every effort to fully address all the recommendations. Below, we outline the suggested areas for improvement along with our corresponding responses and adjustments. We would like to highlight that all modifications are marked in red within the text.
1 - The validated app GerenciaVida arises as a response to the lack of effective systematic instruments for early detection of suicidal behavior. Although it is noted that other apps present low sensitivity and specificity, it would be valuable to include a review of these apps in the format of a comparative table that highlights examples of similar technologies and clarifies how GerenciaVida overcomes these limitations.
Response: A review of technological applications developed for the management of suicidal behavior was included, highlighting their potential, objectives, and limitations. This justified the need for the present study to address unmet needs and to demonstrate the innovative potential of GerenciaVida.
2 - Although the article includes relevant sources such as the WHO and recent post-COVID-19 literature, the theoretical framework on the use of mobile technologies in mental health is limited. Key references on e-health, m-health, and their specific application to suicide prevention are missing. Expanding this section by incorporating critical literature that addresses the ethical dilemmas of using mobile apps in mental health, particularly among vulnerable populations, is recommended. In this regard, we recommend the work of Brazilian researcher Marilia Duque (https://mariliaduque.com/) and the research group Anthropology of Smartphones and Smart Ageing led by British anthropologist Daniel Miller (https://wwwdepts-live.ucl.ac.uk/anthropology/assa/).
Response: References addressing the ethical dilemmas involved in the use of mobile applications in mental health were included.
3 - While broad and addressing clinical, social, and technological aspects, the discussion contains redundancies in the introduction that weaken its analytical depth. It is recommended to restructure this section around three key axes: (1) practical implications of using the app, (2) implementation challenges within public health systems; and (3) sociotechnical or institutional barriers. Additionally, reflecting on the ethical and privacy limitations associated with apps that collect sensitive data would be relevant.
Response: The discussion was revised to follow the suggested structure.
4 - The limitations section is underdeveloped, lacking discussion on key aspects such as the absence of real-world clinical impact measurement and the lack of empirical validation with end users—patients or professionals in actual healthcare settings. It is essential to acknowledge these gaps to provide a more balanced assessment of the study's scope and reliability. As a limitation, it is recommended to include the need for longitudinal studies that assess clinical effectiveness, referral success rates, and prevention of critical episodes.
Response: The discussion was revised to follow the suggested structure.
Best regards,
Reviewer 2 Report
Comments and Suggestions for Authors
This manuscript presents a timely and methodologically sound study on the development and validation of GerenciaVida, a mobile application designed to support healthcare professionals in identifying, classifying, and managing suicidal behavior. The topic is highly relevant to public health and aligns well with IJERPH’s focus on mental health promotion and digital health innovation.
Areas for Improvement
- Validation Scope:
While the study provides strong evidence for content and face validity, it does not assess other psychometric properties such as construct validity, criterion validity, or reliability. Acknowledging this limitation more explicitly and outlining future validation plans would strengthen the manuscript.
- Generalizability:
The expert panel was selected through convenience sampling and is regionally concentrated. The authors should discuss how this may affect the generalizability of the findings to other regions or healthcare systems.
I also wonder about the impact of excluding people with lived experience from the sample.
- Implementation Considerations:
The manuscript would benefit from a discussion of potential barriers to implementation, such as digital literacy, infrastructure limitations, and data privacy concerns, especially given the sensitive nature of suicide-related data.
- Terminology:
Recommend replacing the term “completed suicide” with “death by suicide” or “died by suicide,” as the former may unintentionally imply success or achievement. The latter terms are more neutral, person-centered, and aligned with the current best practices in public health and suicide prevention language. Please revise that language throughout the entirety of the manuscript.
Consider replacing “appearance validity” with the more widely accepted term “face validity” to align with psychometric standards.
- Literature and Reference Integration:
While the manuscript includes a generally current and relevant set of references, there are opportunities to strengthen the literature base by incorporating key international guidelines and frameworks. For example, the inclusion of the World Health Organization’s LIVE LIFE implementation guide (2021) would provide a global context for suicide prevention strategies. Additionally, references to usability and implementation science frameworks—such as the System Usability Scale (SUS), the mHealth Evidence Reporting and Assessment (mERA) checklist, or the person-based approach to intervention development—would enhance the methodological rigor and practical relevance of the study. Incorporating these sources would also help situate the GerenciaVida application within broader digital health evaluation standards and support future scalability and adoption.
Comments on the Quality of English LanguageSome sections would benefit from minor language editing to improve clarity and flow specially, in the discussion section. Many of the paragraphs consist of single sentences, which appear underdeveloped and/or inappropriately split.
Author Response
Dear Reviewer,
We appreciate your careful analysis, the issued review, and your requests for important revisions, which have led to significant improvements in the revised version. We have made every effort to fully address all the recommendations. Below, we outline the suggested areas for improvement along with our corresponding responses and adjustments. We would like to highlight that all modifications are marked in red within the text.
1 - Validation Scope: While the study provides strong evidence for content and face validity, it does not assess other psychometric properties such as construct validity, criterion validity, or reliability. Acknowledging this limitation more explicitly and outlining future validation plans would strengthen the manuscript.
Response: We acknowledge the limitations pointed out and have outlined future validation plans that would strengthen the manuscript.
2 - Generalizability: The expert panel was selected through convenience sampling and is regionally concentrated. The authors should discuss how this may affect the generalizability of the findings to other regions or healthcare systems. I also wonder about the impact of excluding people with lived experience from the sample.
Response: We clarified in the discussion that although a regional concentration of the expert panel is acknowledged—resulting from convenience sampling—the number of participants from other regions of the country is significant. This aligns with established parameters for validation studies, ensures a sample composition above the minimum required, and includes multiple contexts of professional practice. These aspects allow for the gathering of diverse perspectives on the screening, prevention, and management of suicidal behavior and suggest the applicability of the technology in territories with distinct socioeconomic, cultural, and structural dynamics. Including the life perspectives of individuals who have experienced the phenomenon has been incorporated under the technological limitation section.
3 - Implementation Considerations: The manuscript would benefit from a discussion of potential barriers to implementation, such as digital literacy, infrastructure limitations, and data privacy concerns, especially given the sensitive nature of suicide-related data.
Response: A discussion on potential sociotechnical, educational, and institutional barriers and challenges to the implementation of GerenciaVida has been added to the discussion section, which also includes an approach to data privacy, especially considering the sensitive nature of the topic.
4 - Terminology: Recommend replacing the term “completed suicide” with “death by suicide” or “died by suicide,” as the former may unintentionally imply success or achievement. The latter terms are more neutral, person-centered, and aligned with the current best practices in public health and suicide prevention language. Please revise that language throughout the entirety of the manuscript. Consider replacing “appearance validity” with the more widely accepted term “face validity” to align with psychometric standards.
Response: Terminologies have been adjusted as recommended.
5 - Literature and Reference Integration: While the manuscript includes a generally current and relevant set of references, there are opportunities to strengthen the literature base by incorporating key international guidelines and frameworks. For example, the inclusion of the World Health Organization’s LIVE LIFE implementation guide (2021) would provide a global context for suicide prevention strategies. Additionally, references to usability and implementation science frameworks—such as the System Usability Scale (SUS), the mHealth Evidence Reporting and Assessment (mERA) checklist, or the person-based approach to intervention development—would enhance the methodological rigor and practical relevance of the study. Incorporating these sources would also help situate the GerenciaVida application within broader digital health evaluation standards and support future scalability and adoption.
Response: The inclusion of the WHO’s LIVE LIFE Implementation Guide (2021) has been carried out, and the importance of usability evaluation, implementation science, and the person-centered approach has been acknowledged for the development of future studies.
Best regards,